# A goalkeeper's performance in stopping free kicks reduces when the defensive wall blocks their initial view of the ball

**Theofilos Ch. Valkanidis**[1]*, **Cathy M. Craig**[2,3], **Alan Cummins**[1], **Joost C. Dessing**[1]

**1** School of Psychology, Queen's University Belfast, Belfast, Northern Ireland, United Kingdom, **2** School of Psychology, Ulster University, Belfast, Northern Ireland, United Kingdom, **3** INCISIV Ltd., Belfast, Northern Ireland, United Kingdom

* tvalkanidis01@qub.ac.uk

**Data Availability Statement:** All files including figures, videos, data, data analyses are available from the OSF database (https://osf.io/s2bc9/).

## Abstract

Free kicks are an important goal scoring opportunity in football. It is an unwritten rule that the goalkeeper places a wall of defending players with the aim of making scoring harder for the attacking team. However, the defensive wall can occlude the movements of the kicker, as well as the initial part of the ball trajectory. Research on one-handed catching suggests that a ball coming into view later will likely delay movement initiation and possibly affect performance. Here, we used virtual reality to investigate the effect of the visual occlusion of the initial ball trajectory by the wall on the performance of naïve participants and skilled goalkeepers. We showed that movements were initiated significantly later when the wall was present, but not by the same amount as the duration of occlusion (~200ms, versus a movement delay of ~70-90ms); movements were thus initiated sooner after the ball came into view, based on less accumulated information. For both naïve participants and skilled goalkeepers this delayed initiation significantly affected performance (i.e., 3.6cm and 1.5cm larger spatial hand error, respectively, not differing significantly between the groups). These performance reductions were significantly larger for shorter flight times, reaching increased spatial errors of 4.5cm and 2.8cm for both groups, respectively. Further analyses showed that the wall-induced performance reduction did not differ significantly between free kicks with and without sideward curve. The wall influenced early movement biases, but only for free kicks with curve in the same direction as the required movement; these biases were away from the final ball position, thus hampering performance. Our results cannot suggest an all-out removal of the wall–this study only considered one potential downside–but should motivate goalkeepers to continuously evaluate whether placing a wall is their best option. This seems most pertinent when facing expert free kick takers for whom the wall does not act as a block (i.e., whose kicks consistently scale the wall).

## Introduction

It is the fifth minute of the 2018 FIFA World Cup semi-final England-Croatia. Kieran Trippier has just scored England's first free kick goal at a World Cup for 12 years. Prior to the free kick,

**Funding:** The research leading to these results has received funding from the European Union Seventh Framework Programme FP7-CIG under grant agreement n˚ [334202], awarded to Joost C. Dessing, and from the European Research Council under grant agreement n˚ [210007], awarded to Cathy Craig. The work is part of a PhD project that has received funding from the European Union's Horizon 2020 research and innovation programme under the Marie Sklowdowska-Curie grant agreement n˚ 754507. Alan Cummins's contribution was also funded through a project within an Impact Accelerator Award from the UK's Engineering and Physical Sciences Research Council (EP/R511602/1), awarded to Queen's University Belfast. The funders provided support in the form of salaries for authors and purchasing of equipment but did not have any additional role in the study design, data collection and analysis, decision to publish, or preparation of the manuscript. The specific roles of these authors are articulated in the 'author contributions' section.

**Competing interests:** The authors have declared that no competing interests exist. Although Professor Craig is now the CEO of INCISIV Ltd., a commercial entity (founded in May 2018), INCISIV Ltd. has had nothing to do with the design of our study, has not funded it nor is it set to gain financially from the results. None of our results can be commercialized by this company. The affiliation with INCISIV Ltd. does not alter our adherence to PLOS ONE policies on sharing data and materials.

Croatia's goalkeeper Danijel Subašić placed a six-player wall and England added three more, which greatly impacted the goalkeeper by limiting visibility of the striker and initial ball trajectory until the ball passed the wall. Trippier struck the free kick over the wall, perfectly aimed into the top right corner of the goal; Danijel Subašić saw the ball late and did not have enough time to react and reach the ball. This is an example of a situation where, in hindsight, one may wonder whether the goalkeeper was affected by the wall. The negative effects of occluding the initial part of the ball flight using a wall may be apparent to goalkeepers, but their effects have never been scientifically investigated (but see [1]); this study fills this gap. Elite players can shoot free kicks with speeds in excess of 28m/s [2], which means that many free kicks in football take less than a second to reach the goal. Typically, a goalkeeper positions four or more defenders in a wall to block a subset of the possible free kick trajectories. This wall may occlude (part of) the pre-contact kinematics of the kicker and approximately 150-250ms of the ball's initial trajectory. For the time-pressed task of blocking penalty kicks goalkeeper performance depends strongly, if not critically, on early information concerning kicker movements [3–7], with ball flight information only being used for fine-tuning [8, 9], for instance for goalkeeper movement in the horizontal direction [10]. Free kicks allow more time for online control based on ball flight information, which means balls can be stopped even when the kicker is not seen at all, as evident from our previous studies [11, 12]. Lab-based one-handed catching studies with non-expert catchers have shown that early occlusion of the ball trajectory results in later initiation and reduces catching performance [13, 14], due to less information and less time being available to guide the interception. The question is whether these effects generalize to practical scenarios like goalkeeping, which can involve catching with either or both hands. For skilled catchers (i.e., goalkeepers) this is of course a practically relevant question because free kicks represent serious goal scoring opportunities in professional football [15].

Here, we used an immersive interactive virtual goalkeeping simulator to compare goalkeeping performance for the same free kicks with and without a defensive wall. This allowed us to evaluate the effects of occlusion by this wall of the initial ball flight on goalkeeping movements. We set out to test several predictions. Based on the catching research discussed in the preceding paragraph, we expected the wall to result in a later movement initiation and reduced goalkeeping performance. The latter effect should be particularly evident for trajectories with shorter flight times, since catching performance is negatively impacted when less time is available after the ball comes into view [16–18]. Separate tests were conducted for naïve participants and skilled goalkeepers to directly address the generalizability of previous catching research to practical scenarios (i.e., for naïve participants) and the potential practical implications (i.e., for skilled goalkeepers). In addition, we tested the expectation that the performance of skilled goalkeepers would be affected less by the presence of the wall than that of naïve participants, since they would be able to capitalize on their advanced visual [19], motor [20] and visuomotor [21, 22] capabilities once the ball comes into view from behind the wall. Other than this, our study was not designed as a traditional expert-novice paradigm, so no further group comparisons were planned. We did, however, conduct several further exploratory analyses (detailed in the Methods), which all concerned the effect of the wall, not the effect of expertise.

As this study focused uniquely on the occlusion of the initial ball trajectory, our simulator did not include a visual representation of the kicker. The use of simulated, rather than recorded, free kick trajectories provided exact control over the parameters of ball motion, which would be much harder to achieve when showing kicker kinematics (and the resulting, recorded ball trajectory). Our omission of a kicker does not mean we think visual information about movements of the free kick taker is not useful for the goalkeeper; we address this in more detail in the Discussion. We tested the aforementioned effects of the wall across a realistic

range of ball trajectories, which varied in terms of their flight time, spin-induced sideward curve, and the position they entered the goal. We further explored how the effects of visual occlusion by a defensive wall on performance was modulated by these ball flight parameters.

## Methods

### Participant

The experiment involved the participation of 15 naïve participants (age 27.9±6.2 years) and 10 skilled goalkeepers (age 22.4±3.7 years). Naïve participants had no experience with goalkeeping, but some were football club members during their childhood. Our skilled goalkeepers had, prior to the experiment, played at least 5 years continuously as a goalkeeper within the local first amateur division or higher (following [23], these are classified based on skill level and highest level of competition as: 1 expert/International, 2 advanced/National, 2 intermediate/State|Provincial, and 5 basic/Regional|Local). All participants provided written informed consent prior to the experiment.

### Experimental set-up

The study took place in a large laboratory designed to capture human movements of large amplitudes. All experiments were conducted using an HTC VIVE immersive, interactive virtual reality system. This system consists of a head-mounted display (one AMOLED screen per eye, 3.6" diagonal, resolution: 1080 x 1200 pixels; refresh rate: 90 Hz; field of view: 110 degrees), two wireless VIVE controllers and 2 base stations. The VIVE system uses infrared-based motion tracking to monitor movements of the head-mounted display and of two wireless controllers in 6 DoF, which are translated into the virtual world coordinates in real-time. Both (active) cameras (called 'base stations') were located in opposite corners above a 4m x 4m calibrated area (height: 3m, angled 45deg. downward relative to the horizontal plane). The virtual environment was created using Unity 3D 5.5.0. The virtual football pitch dimensions (goal line: 60m touch line: 100m) were compliable with the Law 1 of the IFAB game regulations. A coding error meant that the ball was slightly larger (circumference 0.769m [radius 0.2447m], outside Law 2 of the IFAB regulations [circumference 0.68–0.70m/radius 0.216m-0.223m]) than intended (circumference 0.691m [radius 0.22m], within Law 2 of the IFAB regulations). The environment was oriented relative to the real environment such that the virtual goal line was parallel to but slightly offset, to the diagonal of the 4m x 4m area calibrated with the VIVE system, in order to maximize the lateral range of hand movements covered by the motion tracking system. Participants wore actual match goalkeeping gloves (Sondico, Shirebrook, UK) to which VIVE controllers were attached using Velcro straps. The position and orientation of the virtual hands were offset relative to the controllers to match the actual hand positions in pre-experiment calibrations–no individual optimization of this offset was performed. The VIVE cable from the head-mounted display loosely hung behind the participant, suspended from a ring attached to a wire strung at 3m high, parallel to the virtual goal line (affording free movement without the cable touching the ground). In the virtual environment, the positive x-axis pointed leftward from the goalkeeper's perspective, the positive y-axis upward, and the positive z-axis into the goal. Contact between the virtual ball and either of the hands was defined using Unity's collision detection functionality. In fact, we used invisible square collider objects covering the palm of the virtual hands [0.12m along the proximal-to-distal axis, 0.10m wide, 0.035m thick]).

Free kick trajectories were created using a previously published aerodynamics model [12]—for similar models, see [24–26]—for which the Matlab scripts are made available with this study (https://osf.io/s2bc9/). Note that the simulations (i.e., effects of air friction/spin) were

based on a ball with a 0.22m diameter, which deviated slightly from the used visible ball size (see above). The simulations ended when the ball was around 2m past the goal line. All 3D ball trajectories were saved in separate text files (90Hz) loaded prior to each trial; the associated rotational velocities to be applied to the virtual ball (fixed values per trial) were saved in (and thus loaded from) individual trial sequence files (txt format), which also contained the relevant details for each trial. This loading, as well as the trial sequence, environment physics, sounds, ball motion, wall visibility, and data storage were controlled using C# scripts attached to virtual objects in Unity (project available on request).

Free kicks originated from a distance of 23 meters, exactly in the centre of the goal, which tends to be harder for the goalkeeper to stop [15]. There were 18 unique, experimental ball trajectories: 3 sideward arrival positions (1.5m, 0m, -1.5m [all entering the goal at a height of 1.75m], three levels of sideward curve (leftward, none, rightward), and two flight times (1s and 1.2s, defined as the duration between ball release and the ball centre passing the middle of the goal line). Sideward curve was generated using ball spin. Leftward curve involved counter-clockwise spin, no curve evidently involved no spin, and rightward curve involved clockwise spin. The spin axis was determined as follows: 1) tilt a vertical axis 15deg away from the goal around the x-axis, 2) rotate the resulting axis around the vertical y-axis towards the final lateral ball position (angles: -3.73deg., 0deg., 3.73deg.). Spin rate was set to 1800deg./s. The ball paths used are illustrated in Fig 1 (see S1 Video to see the actual ball motion from inside the goal). To increase the general variation in the conditions such that the entire set of trajectories (and thus the resulting behaviour) is closer to what may be encountered in practice (e.g., in a training session), we also used 45 unique dummy ball trajectories: 5 arrival positions (1.5m, 0.75m, 0m, 0.75m -1.5m), 3 arrival heights (1.75m, 2.12m, 2.49m [n.b. the latter hit the bar and thus did not require any interception]) and 3 sideward curve conditions (leftward, none, rightward, with a 900deg/s spin rate). The flight time for these dummy trajectories was set to 1.1s. Note that the dummy trajectories were not included in the data analysis. Fig 1 highlights a crucial aspect of our trajectories. We standardized spin rate, rather than the level of curve; because the sideward deviation (in m) caused by spin builds up over time, a larger sideward deviation occurred for the longer flight time. Moreover, because we also fixed the height at which the ball entered the goal, longer flight times require higher ball trajectories. This meant the duration of occlusion was in fact shorter for longer flight times, since it largely depended on the instantaneous ball height (see balls in bottom right panel in Fig 1). It should be noted that the constraints on the simulations (e.g., fixing flight time and final height) are not necessarily controlled parameters in actual free kicks; they are a consequence of the kick parameters. All trajectories were presented both with a 5-player defensive wall being present and without. S2 Video illustrates both the viewpoint of the participant during the experiment (also demonstrating the visual occlusion caused by the wall), as well as their positioning and movements in the lab.

## Procedures

Before the experiment, all experimental procedures were explained to the participant in detail and any questions were answered; once satisfied, participants signed a consent form. The experimenter subsequently helped the participant put on the head-mounted display and goalkeeper gloves, and to attach the controllers to the gloves. After the virtual environment was activated, the participant was encouraged to look and walk around for some time to get familiar with the virtual reality experience.

Participants were instructed to start each trial with both virtual hands in view approximately 30cm in front of their face, while standing on the goal line in the centre of the goal; to achieve the latter, they used visual references (e.g., structure in grass, penalty spot, goalposts).

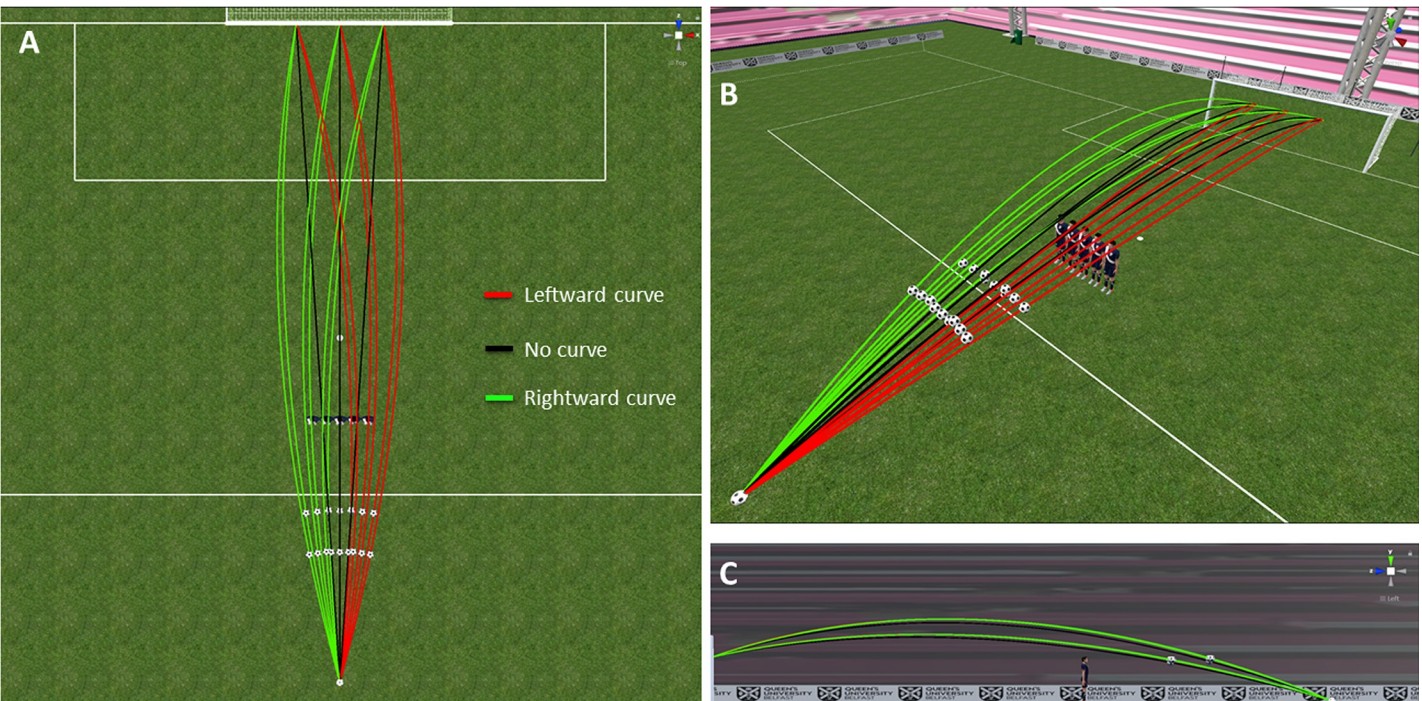

**Fig 1. Visual representation of the ball path in the virtual environment.** A: top view; B: Diagonal side/top view; C: side view (from the goalkeeper's right). Each panel shows all trajectories (see legend in A for colour coding; curve directions expressed from the goalkeeper's perspective), as well as the wall and the ball positions at the frame the entire ball visually cleared the defenders' shoulders. The ball paths are only shown until the goal line. Note that the background (stands) in C was made darker to more clearly show the paths (see B for real colour). See **S1 Video** for moving ball trajectories.

This procedure ensured the hands started approximately within the centre of the goal without using a physical constraint. Every trial started with a sound of a whistle; 1.5sec later a ball-kick sound played, and the ball started moving. Each time the ball collided with another object (hands, ground, nets, goalposts) a unique representation of the physical sound played; details of the collision and subsequent ball motion was controlled by Unity's physics engine. It is important to note that prior to any collision, the ball trajectory was based on the simulated coordinates loaded from the aforementioned text file. After a collision was detected, Unity's physics engine took over the control of the ball's motion, resulting in a realistic representation of the ball bouncing off the collided-with object.

Every participant completed 10 trials for each experimental ball trajectory (5 with a wall, 5 without a wall; total 180 trials) and 4 trials for each dummy ball trajectory (2 with a wall, 2 without a wall; total 180 trials). The experiment started with 20 practice trials, randomly selected from the aforementioned 360 trajectories (note that due to a glitch in our Unity code, the first practice trial was never saved). The order of the 360 trials was randomized for each participant. Breaks were taken after a block of 120 trials. After each trial the 3D position and rotation of the hands, head, and ball were saved in a uniquely named text file. Additional variables such as time, trial number, participant number, trajectory type, wall presence, collision detection and trial information were saved within the same file. All procedures were approved by the Queen's University Belfast School of Psychology Research Ethics Committee (44–2014).

## Data analyses

All anonymized data and analysis code are available at https://osf.io/s2bc9/. In our data analyses, hand position was defined as the centre of the virtual hand; ball position was the centre of

the virtual ball. We defined the origin as the centre of the goal at ground level. In the analyses we flipped the x- and z-axes to suit the authors' preference: the positive x-axis was rightward, the positive y-axis was upward and the positive z-axis was towards the centre of the pitch (all from the goalkeeper's perspective when facing the initial ball position).

As mentioned, we only analysed the 180 experimental trials (i.e., not the dummy trials). Across all data, some trials were excluded for various reasons. To account for frame updates being missed, we required all inter-frame intervals to be .5 and 1.5 times the ideal inter-frame interval (1/90s; 90Hz being the targeted Unity frame rate). To remove trials where the tracking of either of the hands failed temporarily (which would result in a big position jump once tracking was successful again), we required the 3D distance covered by the hands between successive frames not to exceed 25mm (which visual inspection indicated consistently removed erroneous trials). An additional requirement was that these distances should not be ≥5 times larger than the average of the distances for the preceding and next inter-frame intervals (applied separately for each hand). The latter algorithm was optimized by trial-and-error based on visual inspection. To remove incomplete trials (where the experimenter accidentally ended the trial prior to interception), we required the minimal recorded ball position to be 2m from the goal line. For one participant, the three last trials were not saved. On average, 21.9 trials were excluded per participant, using one or more of the aforementioned criteria. We used spline interpolation to reconstruct all 3D positions at exactly 90Hz (aligned with the sample of ball motion onset). Subsequently, hand and head position data were filtered using a 4th order recursive low-pass Butterworth filter (10Hz cut-off) and used to calculate different dependent variables, as explained below.

To test whether presence of the defensive wall influenced movement initiation, we defined movement initiation as the first time either the left or right hand started moving in a lateral direction (calculated with respect to when the ball started to move). For each hand, we calculated the lateral velocity (using *gradient.m*); the hand's moment of initiation was defined as the onset of the first period of at least 200ms during which the absolute lateral velocity exceeded 5% of the hand's maximal absolute lateral velocity between ball motion onset and first sample after the Absolute Error (AE) was determined (see below). Values were averaged across all repetitions, final sideward ball positions, curve conditions, and flight times and compared between conditions with and without a wall, separately for both groups.

To test the prediction that the wall may affect performance, we used AE, that is, how close participants got their hands to the ball. Because this measure captures a magnitude, it is more sensitive/powerful than (and thus preferred over) Success Rate (the percentage of balls touched by either of the hands across the included repetitions for each condition). Success Rate, however, is an intuitive and practically relevant measure that clearly defines the task goal. For this reason, we do report descriptive statistics of this measure of performance alongside all statistics for AE. To limit the number of statistical tests, we did not conduct any statistical analyses for Success Rate, but it showed virtually identical patterns to AE. Although participants were instructed to attempt to block the balls with both hands together, this did not always happen. We thus defined AE based on the hand that got closest to the ball, but also considered the midpoint between the hands as a potential effector to capture situations in which a 2-hand interception was made (and the mid-point between the hands got closer to the ball than either of the hands). We determined when the front of the ball passed each of the three considered effectors (left and right hand and their midpoint) in the forward (z) direction and calculated the distance in the xy plane between the effector and ball at those times; AE was defined as the minimal distance among these. Note that for this calculation, we interpolated ball and hand positions with a step-size of $10^{-5}$s. We averaged AE across all repetitions, final sideward ball positions, curve conditions, and flight times and subsequently compared this variable between conditions with and without a wall, separately for the groups. To test the prediction that

performance would be affected more for shorter flight times, we separately calculated and compared the effect of the wall on performance between the two flight times (1.0s and 1.2s), again separately or both groups. Finally, we tested the prediction that the performance of skilled goalkeepers would be affected less by the wall than that of naïve participants by comparing the effect of the wall on performance (i.e., AE with a wall minus AE without a wall) between the groups.

**The effect of the wall for different free kick parameters.** We explored how the behavioural effects of the wall varied across ball trajectories. We further investigated the modulation of the effect of the wall on AE by flight time (see above) by separately testing the effect of the wall for each flight time. We also followed up the general effect of the wall by comparing it between trials that required body movement (outer final ball positions) and trials that did not (central final ball position), because we noticed extremely high performance for the latter condition, irrespective of the presence of the wall. We tested whether the effect of the wall on performance differed between non-curving and curving ball trajectories. Because for curved ball trajectories interceptive movements (including during free kicks) are biased in the direction of the visual curve and sometimes even initially initiated in the wrong direction [11–13, 26–30], we also examined spatial biases early during the movement. Visual inspection suggested that similar biases were present in our experiment. Because the wall occludes the initial ball trajectory, curve-induced early movement biases may be reduced due to the wall. We defined early movement biases as the sideward head position 500ms before the ball centre passed the goal line, relative to the head position at ball motion onset (n.b., 500ms was used with the aim to sample the head position shortly after the average moment of initiation [note that we did confirm that the patterns reported are not critically dependent on this value, that is, they are virtually identical when using 400ms or 600ms]); this differs slightly from the definition used by in [12], which was motivated from the fact that we could not uniquely determine which hand participants intended to use for interception early during the movement. This analysis ignored trajectories without curve and defined the early movement bias as positive in the direction of the curve (Fig 2A and 2D). The average across the two curve directions was compared between conditions with and without a wall.

Because any effect of the wall in the previous analyses could be beneficial for some ball trajectories (e.g., a larger bias towards the actual final ball position) and detrimental for others (e.g., a larger bias away from the final ball position), we further explored this effect. We tested the effect of the wall on the early movement bias, now defined as positive in the direction of the final sideward ball position, separately for trials with curve in the direction of the final ball position (i.e., congruence between direction of curve and require movement direction) and trials with curve in the direction opposite to the final ball position (i.e., incongruence between direction of curve and require movement direction) (see Fig 2B, 2C, 2E and 2F). For these analyses, $X_{early}$ was averaged across flight times. Because early biases may affect performance, we also compared the difference in the wall-induced performance reduction between congruent and incongruent trials.

**Statistics.** Parametric statistics were used for the moment of initiation, because Shapiro-Wilk tests suggested that the data came from a normal distribution ($p > 0.05$). Non-parametric statistics were used for AE, because a priori, we expected AE to be non-normally distributed due to its positive definite value and optimal central tendency at 0. Parametric statistics were used for all tests involving the early movement bias, because Shapiro-Wilk tests confirmed the data was normally distributed ($p > 0.05$). For both groups, we conducted the three hypothesis tests concerning the effect of the wall and we also compared AE between the groups; we thus adjusted the significance levels for 7 comparisons using a step-down Holm-Šídák procedure (retaining $\alpha = 0.05$ across these tests). Our additional exploratory analyses involved 8 tests for each group, and we thus adjusted the significance levels of these tests for 16 comparisons using a step-down Holm-Šídák procedure (retaining $\alpha = 0.05$ across these tests).

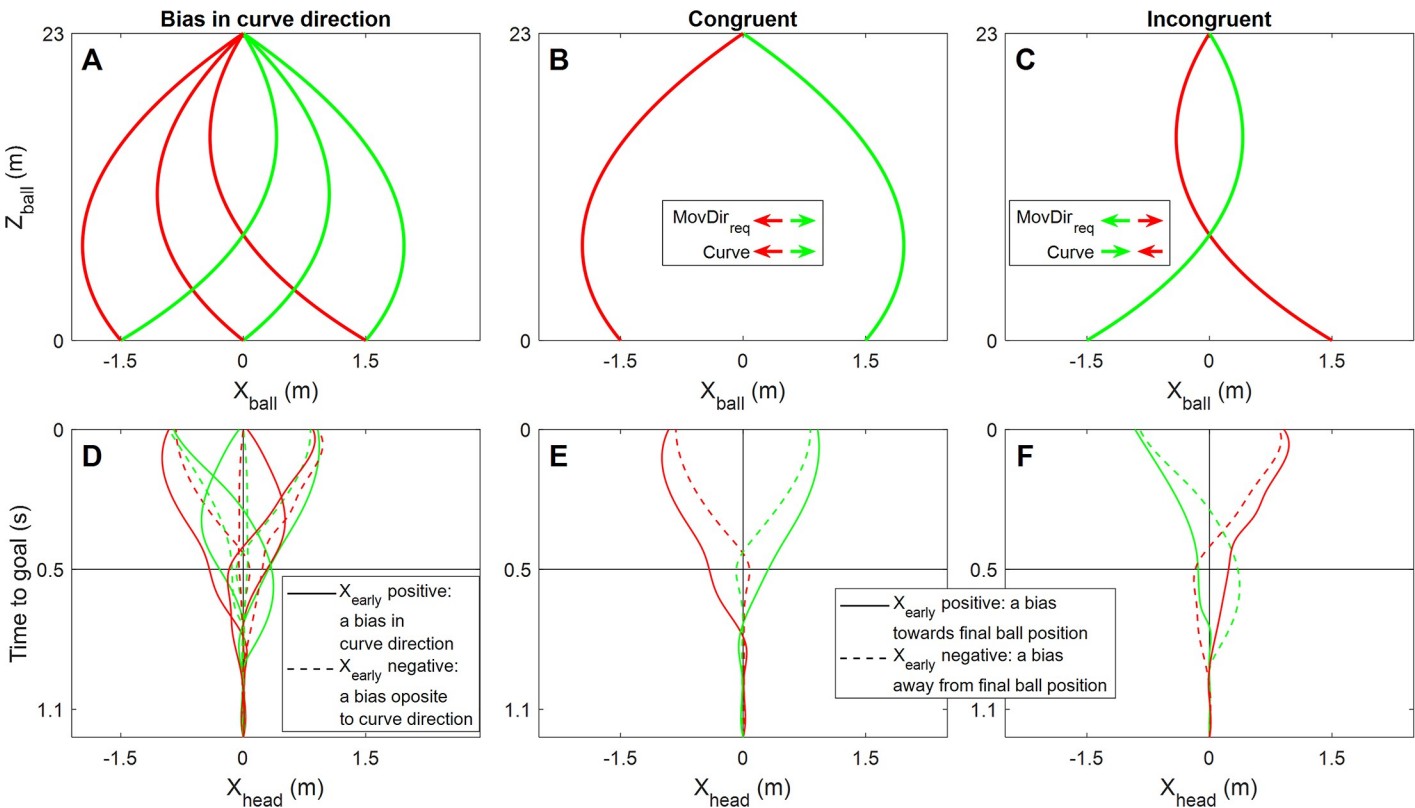

**Fig 2. The definition of early movement biases in two of the follow-up analyses.** For the analysis of early movement bias ($X_{early}$) in the direction of curve all curved ball trajectories were included (A–here illustrated for a flight time of 1.2s) and $X_{early}$ was defined to be positive if the head displacement between the ball motion onset and 500ms before the ball reached the goal line (see black horizontal lines) was in the same direction as the curve (examples in D). For the analysis of $X_{early}$ as a function of the congruence between the directions of requirement movement and curve (as illustrated in panels B [congruent trajectories] and C [incongruent trajectories]) the central final ball position was excluded and $X_{early}$ was defined to be positive if the head displacement between the ball motion onset and 500ms before the ball reached the goal line (see black horizontal lines) was in the same direction as the required movement (towards the final ball position; examples in E and F). In all panels, ball/hand/head trajectories for leftward curve are red and those for rightward curve are green; all trajectories in panels D-F are artificial (i.e., not real data).

## Results

This study involved a virtual reality goalkeeping paradigm, in which participants were standing in the centre of the goal and moved their body and hands predominantly in a lateral direction to block approaching footballs. Lateral hand movements of a representative naïve participant are depicted in Fig 3 for all conditions in this experiment. The trajectories in the figure show that the hands were moved towards the ball; they were influenced by the curve and final position of the free kick. It should be noted that the presence of the wall–which is the key manipulation in this study–expectedly did not qualitatively alter the trajectories and only had a relatively small effect that cannot be appreciated from individual trajectories. We therefore did not use different line styles for the wall conditions in the figure. The wall did, however, significantly affect the movements of both groups, as we discuss in the following section.

### Predicted effects of the wall

Our *first prediction* was that, because the wall causes the ball to come into view later, it should result in a later movement initiation time. Indeed, naïve participants significantly delayed their movement initiation by 86ms on average ($t(14)$ = -12.99, $p_{HS}$ = 2.4·10$^{-8}$, Fig 4A), and skilled goalkeepers by 70ms ($t(9)$ = -5.18, $p_{HS}$ = 0.0035, Fig 4B). Because less of the ball

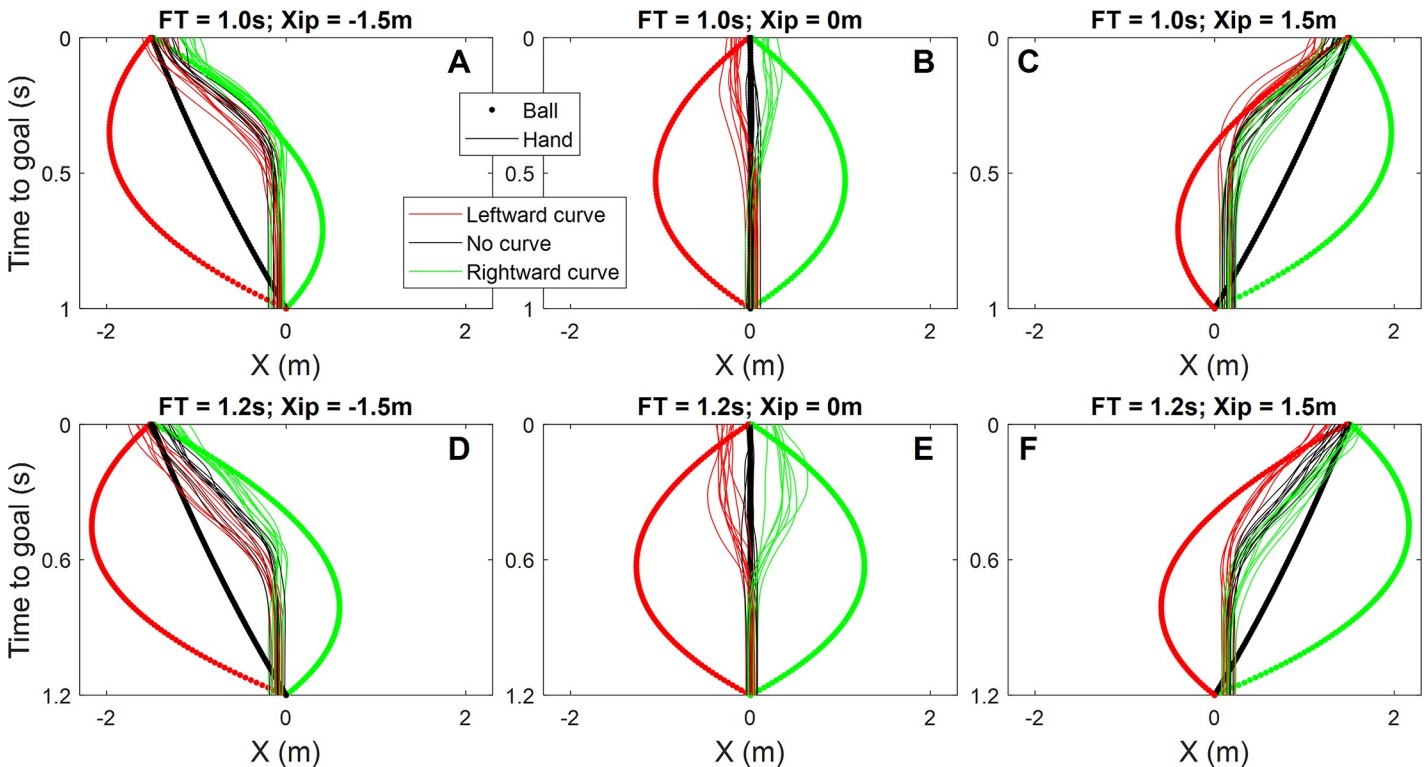

**Fig 3. Representative horizontal hand and ball trajectories.** Ball trajectories with different levels of curvature are represented using bold lines of different colours (red = rightward curve; black = no curve; green = leftward curve) and hand movements are represented using fine lines with the same corresponding colour to match the ball trajectory curvature. The panels depict the lateral ball and hand position as a function of the time before the ball reaches the goal, in the three final positions (Xip = -1.5m[left panels], 0m[middle panels], 1.5m[right panels]). Top panels show positions for 1s ball flight time (FT) and bottom panels show positions for 1.2s ball flight time. Note: left panels show Left hand trajectories, middle panels show the average of Left/Right hand trajectories, Right panels show Right hand trajectories; these were not necessarily the hands used to determine performance (see text).

trajectory is seen by the goalkeepers with the wall, our *second prediction* was that the wall negatively affects performance (which we quantified by the absolute interception error, AE; see Methods). Our data matched this prediction; for naïve participants, this effect had a median magnitude of 3.6cm ($W$ = 94, $p_{HS}$ = 0.027, success rates: No Wall: 74.5%, Wall: 66.4%, Fig 4C), and for skilled goalkeepers 1.5cm ($W$ = 51, $p_{HS}$ = 0.020, success rates No Wall: 85.8%, Wall: 79.1%, Fig 4D). Our *third prediction* was that the wall should have a larger negative effect on performance if less time is available. Indeed, our results revealed a larger wall-induced performance decrement for shorter flight times for both naïve participants ($W$ = -80, $p_{HS}$ = 0.043, median difference 2.9cm, Fig 4E) and skilled goalkeepers ($W$ = -51, $p_{HS}$ = 0.020, median difference 3.1cm, Fig 4F; success rates reported in Table 1). Thus, all predicted effects of the wall were confirmed for both groups. Although the average size of the effect of the wall on AE was more than two times smaller for skilled goalkeepers, this difference did not reach significance in our one test of expertise effects (U = 64, $p$ = 0.56).

### Exploring the effect of the wall for different aspects of the free kick

In addition to the reported hypothesis tests, we explored how other features of the free kick influenced the effect of the wall on goalkeeper movements; these analyses are reported next.

**Effect of flight time.** We tested the effect of the wall on performance (i.e., AE) separately for the two flight times. For both groups the presence of the wall led to significantly greater

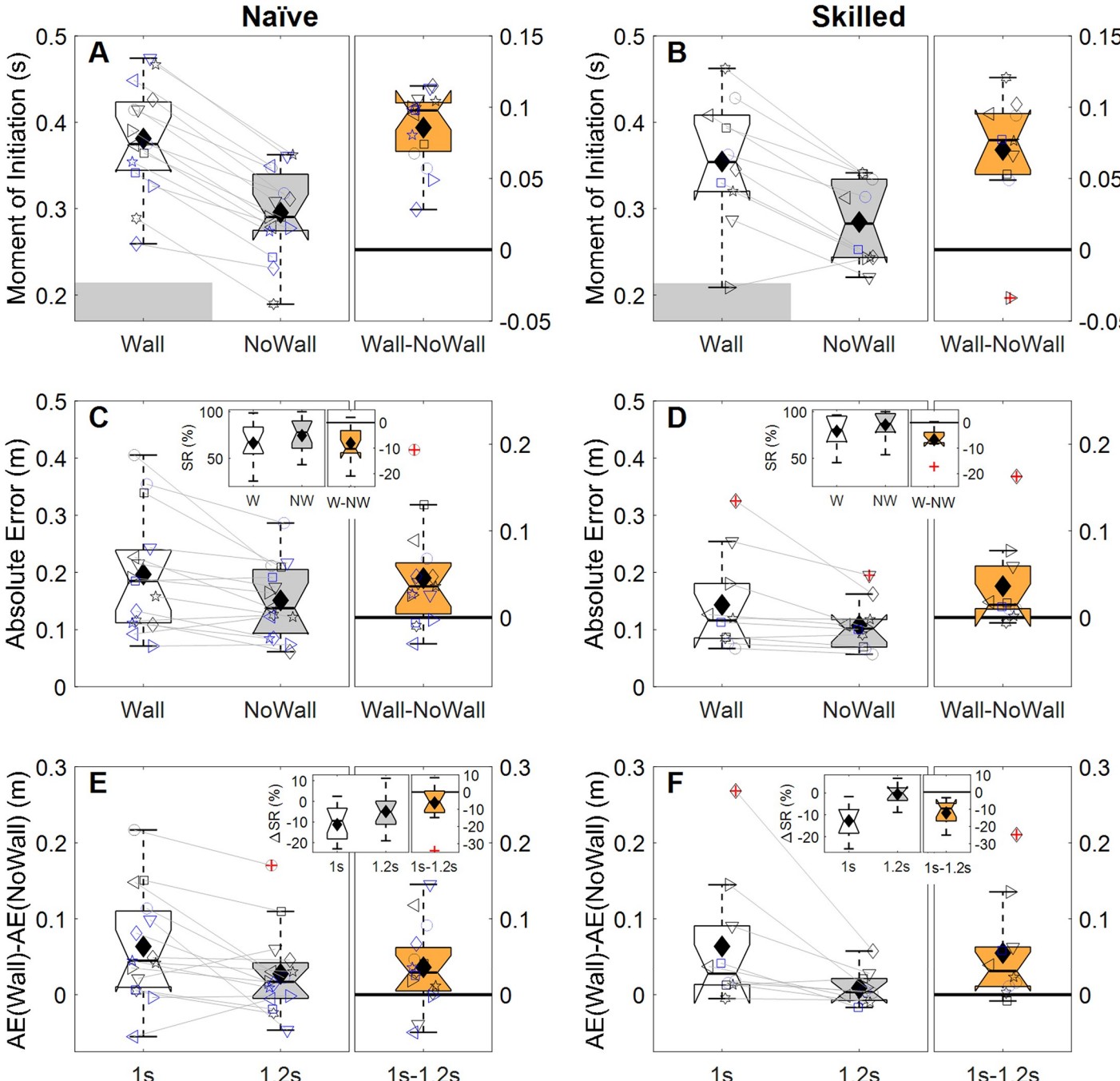

**Fig 4. Hypothesis tests for naïve participants (left panels) and skilled goalkeepers (right panels).** Panels A and B show the moment of initiation (relative to the moment the ball started moving) averaged across conditions with (white box, "Wall") and without the wall (grey box, "NoWall"), as well as their difference (orange box, axis on the right). The grey shaded area shows when the wall occludes the ball. Panels C and D show the absolute error averaged across conditions with (white box, "Wall") and without the wall (grey box, "NoWall"), as well as their difference (orange box, axis on the right). The insets show the success rate (SR in %) in an identical format (W = Wall, NW = NoWall). Panels E and F show the effect of the wall on absolute error (AE(Wall)-AE(NoWall)) averaged across conditions with a 1.0s flight time (white box) and a 1.2s flight time (grey box), as well as their difference (orange box, axis on the right). The insets show the effect of the wall on success rates (ΔSR in %) in an identical format, although for the difference values (orange boxes, 1s-1.2s) the axis is shown on the right. For all boxes, default settings in *boxplot.m* were used (i.e., boxes represent the 25–75 percentiles, whiskers extend to the most extreme data values (excluding outliers [red +s], positioned more than 1.5 times the interquartile range from the box's outer edges, the horizontal black line in the box represents the median, notches represent the 95% confidence interval of the median; in addition, the mean is shown as a black diamond). In the main panels, individual data is depicted using consistent symbol/colour/jitter combinations; for clarity, individual means for the conditions with and without a wall are connected using a grey line.

**Table 1. Success rates on the effect of the wall for different aspects of the free kick.**

| | | Naïve participants | | Skilled goalkeepers | |
|---|---|---|---|---|---|
| | | *No Wall* | *Wall* | *No Wall* | *Wall* |
| **Flight time** | **1s** | 69.3% | 58.1% | 81.5% | 68.7% |
| | **1.2s** | 79.8% | 74.9% | 90.2% | 89.6% |
| **Required movement amplitude** | **0m** | 92.6% | 93.3% | 96.8% | 97.3% |
| | **1.5m** | 65.5% | 53.0% | 80.1% | 70.3% |
| **Trajectory shape** | **No Curve** | 81.9% | 72.2% | 90.8% | 83.0% |
| | **Curve** | 70.9% | 63.5% | 83.3% | 77.2% |
| **Direction curve and required movement** | **Congruent** | 79.6% | 66.1% | 91.6% | 75.4% |
| | **Incongruent** | 42.6%, | 33.5% | 62.5% | 60.5% |

Note: no statistical analyses were conducted for Success Rate (see Methods).

errors for shorter flight times (naïve: $W = 100$, $p_{HS} = 0.022$, median: 4.5cm, skilled: $W = 53$, $p_{HS} = 0.035$, median: 2.8cm), but not for the longer flight times (naïve: $W = 62$, $p_{HS} = 0.46$, median: 1.7cm, skilled: $W = 19$, $p_{HS} = 0.76$, median: 0.3cm, see Table 1 for success rates).

**Effect of required body displacement.** We tested whether the effect of the wall on performance was larger if sideward movement was required. We thus compared the effect of the wall on AE averaged for balls entering the goal at -1.5m and 1.5m with the effect for balls entering the goal in the centre. The wall reduced performance significantly more when sideward movement was required (naïve: $W = 100$, $p_{HS} = 0.027$, median difference: 6.8cm; skilled: $W = 51$, $p_{HS} = 0.046$, median difference: 1.7cm, see Fig 5 and Table 1 for success rates).

**Effects of curve.** We tested whether the effect of the wall on performance differed between non-curved and curved free kick trajectories, but did not find any evidence for such difference for both naïve participants ($W = 20$, $p_{HS} = 0.60$, median: 0.1cm) and skilled goalkeepers ($W = 15$, $p_{HS} = 0.74$, median: 0.4cm) (see Table 1 for success rates). This result, of course, does not mean that curve did not influence goalkeeper movements. Because the wall occludes the

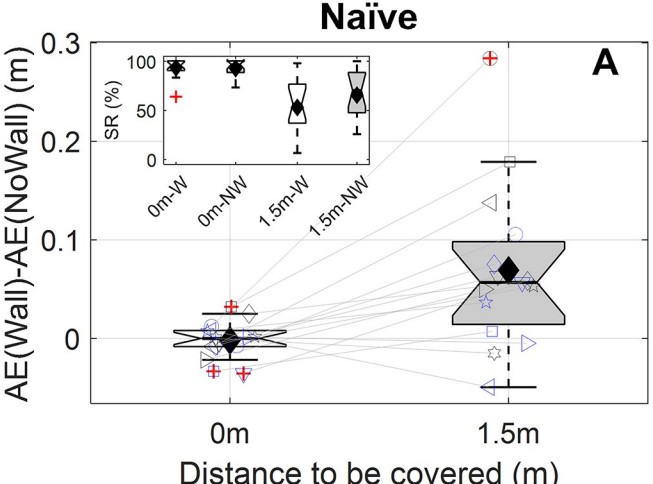
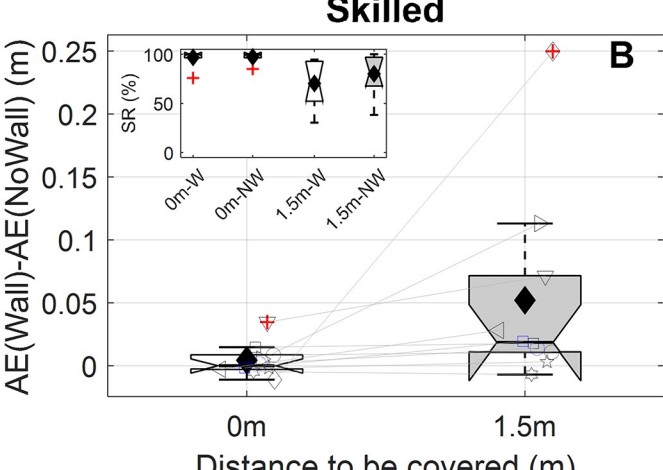

**Fig 5.** Effect of required movement distance on how the wall affects AE for naïve participants (A) and skilled goalkeepers (B). The difference between the absolute error (AE) averaged for conditions with a wall and that averaged for conditions without a wall is shown for the central final ball position (white box, 0m distance to be covered) and averaged across the outer final ball positions (grey box, 1.5m distance to be covered). Formatting of individual data is identical to that described for Fig 4. Insets show the effect of the wall on success rate (SR in %) for conditions without required movement with a wall (0m-W) and without (0m-NW) and conditions with 1.5m required movement with (1.5m-W) and without a wall (1.5m-NW). In the insets individual data is not shown to avoid crowding.

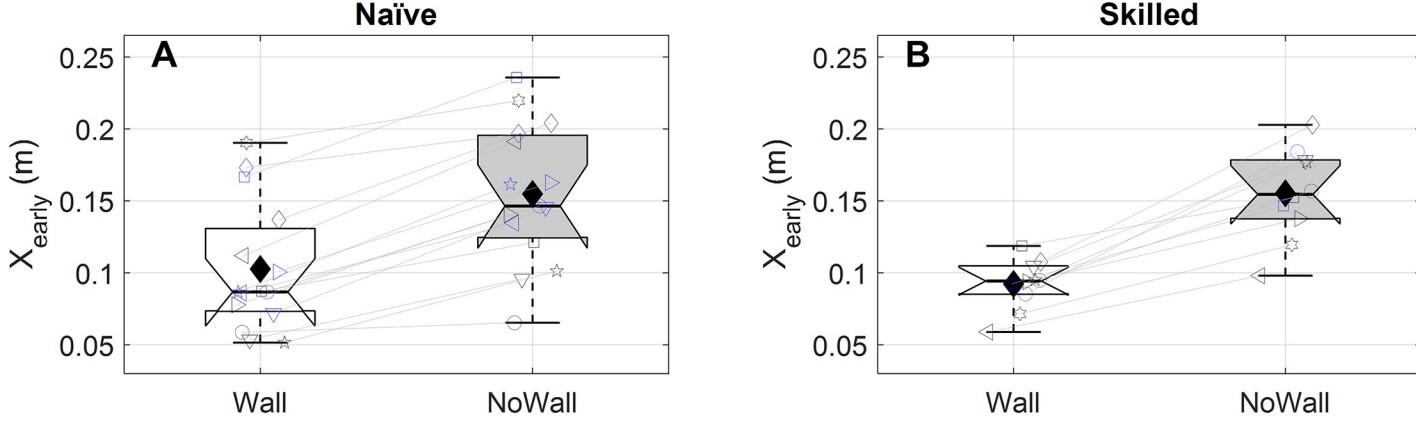

**Fig 6.** Effect of the wall on the curve-induced early movement bias ($X_{early}$) for naïve participants (A) and skilled goalkeepers (B). The early movement bias was defined positive in the direction of curve and subsequently averaged across conditions with leftward and rightward curve, separately for conditions with (white box) and without a wall (grey box). Formatting of individual data is identical to that described for Fig 4.

initial ball trajectory and results in a later initiation, the typically reported curve-induced early movement biases [12] may be reduced due to the wall. We redefined $X_{early}$ as being positive in the direction of the curve (i.e., ignoring trajectories without curve) and compared the average $X_{early}$ between Wall and No Wall conditions. Both groups showed a smaller early sideward bias in the direction of the curve with a wall, compared to without (naïve: $t(14) = 9.48$, $p_{HS} = 2.9 \cdot 10^{-6}$, average 6.0cm reduction, Fig 6A; skilled: $t(9) = 8.54$, $p_{HS} = 1.7 \cdot 10^{-4}$, average 5.8cm reduction, Fig 6B).

**Combined effects of directions of required movement and curve.** We examined whether the aforementioned effect of curve mainly had positive or negative consequences (i.e., reducing or increasing the to-be-covered distance; see Methods), by testing the effect of the wall on the early bias separately for trials with congruent directions of curve and required movement and trials with incongruence thereof. For both groups, we found negative effects of the wall on the early bias for congruent trials (naïve: $t(14) = 9.12$, $p_{HS} = 4.3 \cdot 10^{-6}$, Fig 7A, average effect of the wall: -8.9cm; skilled: $t(9) = 13.15$ $p_{HS} = 4.9 \cdot 10^{-6}$, average effect of the wall: -10.0cm, Fig 7B, see Table 1 for success rates). Both groups failed to show a significant effect of the wall on $X_{early}$ for incongruent trials (naïve: $t(14) = -1.06$, $p_{HS} = 0.77$, Fig 7A, average effect of the wall: 2.1cm; skilled: $t(9) = -1.8$ $p_{HS} = 0.43$, average effect of the wall: 1.3cm, Fig 7B, see Table 1 for success rates). The aforementioned negative effect of the wall on performance and the discussed pattern in the early biases motivated us to examine the difference in the wall's effect on final performance (i.e., AE) between the described congruent and incongruent trials. For naïve participants the pattern for AE appeared to follow the early bias: the wall had a more negative effect on performance for congruent than incongruent trials ($W = -114$, $p_{HS} = 0.0037$, median difference 6.4cm, Fig 7C and see Table 1 for success rates). For skilled goalkeepers, however, AE did not differ between the congruent and incongruent trials ($W = -35$, $p_{HS} = 0.41$, median difference 2.3cm, Fig 7D and see Table 1 for success rates). Irrespective of the effects of the wall, the success rates do highlight that free kicks with incongruent directions of required movement and curve were the more difficult ones to block in general.

## Discussion

This study examined whether visual occlusion of the early part of a ball trajectory caused by the defensive wall affects goalkeepers' performance when moving to stop free kicks. Because

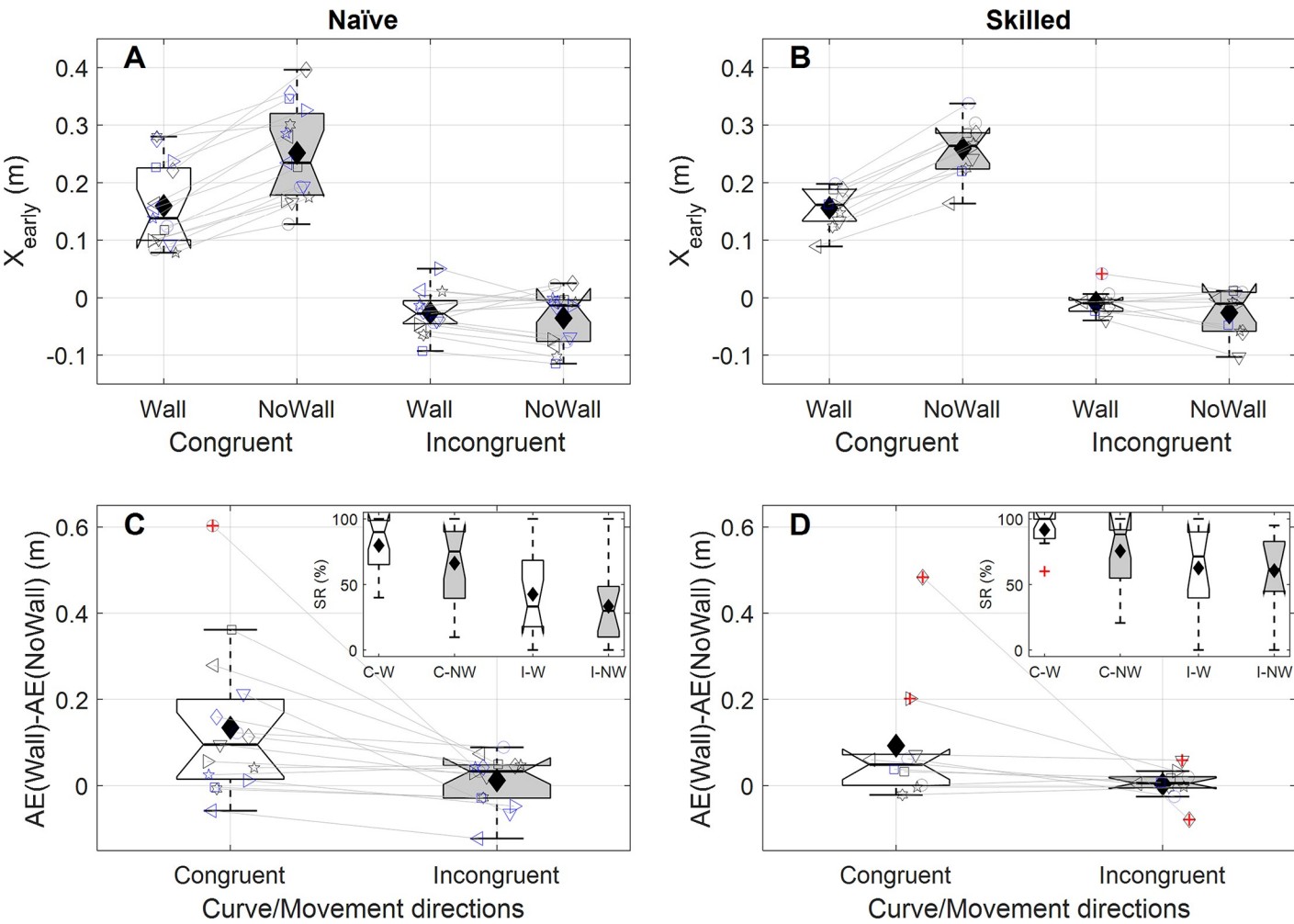

**Fig 7. Effect of congruence of directions of curve and required movement on how the wall influenced the interception movement.** Top panels (A [Naïve participants], B [Skilled goalkeepers]) show the effect of the wall on early movement bias ($X_{early}$, defined positive in the direction of required movement) and bottom panels (C [Naïve participants], D [Skilled goalkeepers]) show the effect of the wall on the absolute error, AE; in all panels the values are averaged across conditions where ball trajectories curved in the direction of the requirement movement (white panels, Congruent) or in opposite direction (grey panels, Incongruent). Formatting of individual data is identical to that described for Fig 4. In panels C and D, insets show the success rate (SR) for congruent trials with (white box, C-W) and without a wall (grey box, C-NW) and for incongruent trials with (white box, I-W) and without a wall (grey box, I-NW).

the wall causes the ball to come into view later, we predicted a delayed movement onset [13–14]. Since this leaves less time to complete the interception and because the wall occludes the initial ball motion, we expected the wall to reduce performance and particularly so for shorter flight times [16–18]. These predictions were confirmed by our experiments. However, we did not find evidence for the prediction that the effect of the wall would be smaller for skilled goalkeepers than for naïve participants. Below we discuss the results in more detail, as well as their theoretical and practical implications.

Both groups of participants delayed their movement onset when the wall was present. Although the occlusion lasted ~200ms, the delay was in the order of 70-90ms, implying that movements were initiated sooner after the ball came into view (and were thus based on less accumulated visual information). This, combined with the shorter time to intercept, resulted in 1.5–3.6cm increase in the spatial error (7–8% fewer interceptions), an effect that–contrary

to our expectations–did not differ significantly between our skill groups. Performance reductions were even more extreme for shorter flight times (2.8–4.5cm larger spatial error / 11–13% fewer interceptions) and for free kicks requiring the goalkeeper to move in a sideward direction (1.9–5.7cm larger spatial error / 10–13% fewer interceptions). Further effects of the wall were observed on the early biases within the movement. Early movement biases in the direction of curve have been previously reported [11–13, 26–30] and we found that these biases were reduced when the wall was present. Since reduced curve-induced biases would be positive if the curve direction is opposite to the required movement direction and negative if the curve is in the same direction as the required movement, we separately tested the effect of the wall on the early bias for these subsets of conditions. We only found evidence for a negative influence of the wall on the early movement bias for congruent trajectories (i.e., away from the final ball position). In naïve participants, these early biases ultimately influenced interception performance for these trajectories; our skilled goalkeepers showed a similar pattern for the early biases, but no significant biases were found in their interception performance for congruent trajectories. Thus, the early movement is affected more consistently by the wall for congruent trials and, tentatively, only skilled goalkeepers were able to correct for this during their movement. In the following, we discuss these findings in terms of the visual control of movement.

It is interesting to consider the visuomotor mechanisms underlying the effects of the wall on goalkeeping behaviour. Evidently, our results only pertain to visual control related to ball motion, since an avatar who kicks the ball was not used in our simulator (but see below). As opposed to tasks relying heavily on visual anticipation, where the early visual information (about an opponent's movement kinematics and ball flight) and situational probabilities are critical [8, 9], in free kicks there is enough time for movement corrections based on visual information about ball motion. But our results show that even missing the early bit of this information about ball motion is detrimental to performance. The ball came into view later because of the wall; the resulting later movement initiation left less time to successfully perform the interception. It is known that for interceptive movements (if enough time is available), a later initiation time influences the entire movement [12, 29], something we also observed in the patterns of the early movement bias, predominantly for trajectories curving in the direction of the required movement. This may be due to time constraints, namely the same distance needing to be covered in less time. The role of time constraints (i.e., the time available for vision-based movement control [16–18]) is supported by our finding that the wall mainly affects performance for short flight times. Kinematic changes could also be related to a different history of visual information pick up. At the moment the ball comes into view from behind the wall, without a wall the goalkeeper would have already had ~200ms of information concerning the ball trajectory (on top of more accurate kinematic information concerning the kicker, see below). Evidently, having better/more information can improve movement control and ultimately performance (as we observed). The relative contributions of increased time-pressure and reduced accumulation of information to the detailed effects of the wall needs to be addressed in future experiments.

The S1 File presents two additional experiments–including only naïve participants–conducted to test the effects of the wall for a wider range of conditions (i.e., different initial distances and different initial sideward positions). These experiments mostly replicated the effects of the wall on initiation and performance, although the performance reduction just failed to reach significance for different sideward initial ball positions. Although our results, as well as previous findings [12], suggest movements of skilled goalkeepers are influenced by the visual information of ball trajectories (or their absence, in our experiment), in a similar way, the lack of skilled goalkeepers in the additional experiments means that generalizability for this group awaits empirical verification.

### Practical implications

Our results do have practical relevance, but it must be emphasized that, even though our virtual reality goalkeeping simulator was realistic, some aspects differed from real goalkeeping. It is appropriate to first evaluate the potential effects of these deviations on our findings and their practical relevance. Our focus on occlusion of the initial ball trajectory meant our desire to control over the parameters of ball motion took precedence over including a kicker avatar in the scene. It is nevertheless likely that goalkeepers also benefit from information about the kicker movements [8, 9, 31]. During free kicks, aspects of kicker kinematics could inform the goalkeeper on the expected direction of curve (e.g., any curve will most likely be in the direction of the kicking foot) and possibly the required direction of movement (see [32] for comparable information use in tennis). In fast interceptive movements, kinematic information about the opponent results in an earlier initiation and better performance [31]. Because the wall would occlude the kicker in our task, any such performance benefit would be larger (or even unique to) the condition without the wall. By implication, our results may in fact underestimate the negative effects of the wall; thus, although the absence of a kicker deviates from reality, we think this does not impact our conclusions concerning the effects of occlusion in practice.

Several situational constraints were implemented in our task to maximize experimental control. Firstly, the wall and goalkeeper were positioned centrally, while in practice goalkeepers often opt to place the wall off centre to block off one side of the goal, while biasing their initial position slightly away from the blocked-off side to capitalize on situational probabilities [33] and increase the view of the kicker's movements. Evidently, the drawback of biasing is that it leaves one side of the goal "exposed" if the ball makes it over the wall. Besides the fact that a single, central wall position requires fewer conditions/trials (compared to left and right off- centre placements), our key motivation for these constraints was that all free kicks could be stopped using a step and reach, which removes a key benefit of biasing the initial position. For a central initial ball position, there would be limited possibilities for increasing the view of the kicker, since the final kicker movement (i.e., foot-ball contact) can only be seen for rather extreme and thus unpractical sideward biases (e.g., ~2.5m for free kicks from 30m). The exact effects of allowing biased wall and goalkeeper positioning can only be ascertained when our task can be safely completed with diving (as changes in initial position might increase the required movement distance). The arrival of wireless virtual reality will allow us to test this safely and experimentally establish the benefits/downsides of allowing goalkeepers to position the wall and themselves at a position of their choice. Allowing dives could also impact our findings. It is telling that we found effects of occlusion for free kicks considerably less challenging (in terms of the required sideward displacement) than those encountered by goalkeepers in games (mostly dealt with by diving). Indeed, we expect performance to be affected even more by the wall for such more challenging free kicks, since a delayed initiation would be more problematic if a larger distance must be covered (see also [13]). This also remains to be established in future investigations.

Although the reported effects were very consistent and may well be underestimated (see preceding discussion), we would certainly not suggest goalkeepers should *never* place a wall. However, our findings should motivate goalkeepers and trainers to evaluate whether to place a wall or not. Our results do suggest that such an evaluation is most relevant when faced with free kick takers known to shoot hard (leaving the goalkeeper little time). Evidently, the opponent would need to consistently shoot challenging shots that scale the wall (i.e., a free kick expert), when there is potential for the negative effects of (occlusion by) the wall to outweigh the potential benefit of blocking shots. It would be a case of comparing the percentage of shots

into the wall with the performance improvement due to seeing the entire ball flight. The first may be estimated from historic figures for each opponent, while the latter requires field tests; there may well be a role for virtual reality performance testing in this respect in the future. Indeed, our study further paves the way for in-practice applications of virtual reality; standardized performance tests for instance could inform training scenarios (i.e., practicing those scenarios where individual performance is challenged).

The wall or its removal may also affect free kick takers in two ways. In absence of the wall, some kickers may be able to deliver a more powerful shot–possibly traveling through the spatial window that would have been occupied by the wall. This would be a beneficial effect if the more powerful shot is more challenging for the goalkeeper. The consistent use of a wall suggests that teams believe that the latter would always outweigh any benefits of removing the wall. It has been reported anecdotally that the wall (i.e., the visual wall-goal configuration) may act as a reference for the kicker's aim, with shots aimed over a specific player's heads [34]. Removing this reference could affect the accuracy of the shot, and thus negatively affect the kicker. For both these effects, explicit (scientific) evaluations have not yet been undertaken, so a cost-benefit analysis is not possible at this stage.

Our findings confirm assumptions of expert free kick takers, who have been exploiting occlusion in free kick scenarios over the years. Attacking teams have frequently placed players next to the wall (as in the Trippier example mentioned in the Introduction), which shows the players understand the potential benefits of obstructing the goalkeeper's view. The current rules (IFAB Laws of the Game 2019–20, Law 13) state that these players must stand at least 1m from the wall (if there is a wall of three or more defenders). If these players are positioned closer to the free kick location (i.e., at 8.15m) increased occlusion can be achieved within the regulations. However, these players can be placed offside by the defending team, since players blocking the opponent's line of vision are actively involved in play (IFAB Laws of the Game 2019–20, Law 11). If the goalkeeper would decide to leave out the wall altogether, the attacking team could place their own wall (to generate occlusion and possibly help the kicker's aim–a strategy used in the past by Dutch free kick expert Pierre van Hooijdonk [35]). However, this wall again can be placed offside by the defending team. Clearly, wall-induced occlusion renders the free kick scenario a game of cat and mouse between the attacking and defending teams.

## Conclusion

In conclusion, we used a virtual reality goalkeeping simulator to show conclusively that visual occlusion of the initial ball trajectory by the wall in free kick scenarios in football negatively affects goalkeeping performance. This is an example of a situation for which lab-based effects of visual occlusion generalize to practical scenarios. The effect was substantial and consistent enough to warrant the suggestion that goalkeepers should evaluate whether placing a wall is always their best option. For instance, when faced with a free kick expert whose shots on goal are hardly ever blocked by the wall, the only possible goalkeeping benefit of the wall would be preventing some more challenging shots. Scientifically, the relative contributions of various effects of the wall on the goalkeeper (e.g., occlusion of the kicker and initial ball trajectory) and kicker (i.e., aiming benefits and kicking strategies) remain to be determined–such knowledge evidently is also highly relevant for practical impact.

## Supporting information

**S1 File. Methods and results for Experiments 2 and 3.** This file presents the methods and results of two additional experiments with naïve participants, in which the initial distance

(Experiment 2) and initial sideward position (Experiment 3) were varied.
(PDF)

**S1 Video. Dynamic ball trajectories with different flight times–occlusion by the wall.** This video illustrates the virtual environment and the trajectories used in the experiment. The colour of the lines matches the trajectory type described in the Methods section. The two top left boxes visualise the goalkeepers view with and without wall through the fixed head mounted display. The bottom left box presents a 3D view behind and above the virtual goalpost, the top right a 2D top view and the bottom right a 2D side view.
(MP4)

**S2 Video. Participant view and behaviour in the virtual goalkeeping simulator.** The video illustrates the behaviour in a sample of trials used for this study. The left screen shows the virtual environment through the participant's continuously changing perspective and right screen presents the corresponding real movements of a participant (note: this is not actual data recorded during the experiment). The same visualizations are shown for the experiment presented in the manuscript (called Experiment 1) and for those presented in the S1 File (called Experiments 2 and 3). Before the visualisation for each experiment, an explanatory slide is shown containing the 2D top view (left) and the conditions used (right). Note that a higher resolution, higher frame rate version of this video is available at https://osf.io/fhq76/.
(MP4)

**S3 Video. Dynamic ball trajectories with different initial distances–occlusion by the wall.** This video illustrates the virtual environment and the trajectories used in the first experiment presented in the S1 File (with variations in initial ball distances). The colour of the lines matches the trajectory type described in the Methods section. The two top left boxes visualise the goalkeepers view with and without wall through the fixed head mounted display. The bottom left box presents a 3D view behind and above the virtual goalpost, the top right a 2D top view and the bottom right a 2D side view.
(MP4)

**S4 Video. Dynamic ball trajectories with different initial sideward positions–occlusion by the wall.** This video illustrates the virtual environment and the trajectories used in the second experiment presented in the S1 File (with variations in initial sideward ball positions). The colour of the lines matches the trajectory type described in the Methods section. The two top left boxes visualise the goalkeepers view with and without wall through the fixed head mounted display. The bottom left box presents a 3D view behind and above the virtual goalpost, the top right a 2D top view and the bottom right a 2D side view.
(MP4)

## Acknowledgments

The authors thank Oisin McManus, Callum Kane, Caroline Wray, Emma McAlinden, and Mark Poland for their help in collecting part of the data for Experiments 2 and 3, presented in the S1 File.

## Author Contributions

**Conceptualization:** Theofilos Ch. Valkanidis, Cathy M. Craig, Joost C. Dessing.

**Data curation:** Theofilos Ch. Valkanidis, Joost C. Dessing.

**Formal analysis:** Theofilos Ch. Valkanidis, Joost C. Dessing.

**Funding acquisition:** Cathy M. Craig, Joost C. Dessing.

**Investigation:** Theofilos Ch. Valkanidis.

**Methodology:** Theofilos Ch. Valkanidis, Joost C. Dessing.

**Project administration:** Theofilos Ch. Valkanidis, Cathy M. Craig, Joost C. Dessing.

**Resources:** Theofilos Ch. Valkanidis, Cathy M. Craig, Joost C. Dessing.

**Software:** Theofilos Ch. Valkanidis, Alan Cummins, Joost C. Dessing.

**Supervision:** Theofilos Ch. Valkanidis, Cathy M. Craig, Joost C. Dessing.

**Validation:** Theofilos Ch. Valkanidis, Alan Cummins, Joost C. Dessing.

**Visualization:** Theofilos Ch. Valkanidis, Joost C. Dessing.

**Writing – original draft:** Theofilos Ch. Valkanidis.

**Writing – review & editing:** Theofilos Ch. Valkanidis, Cathy M. Craig, Alan Cummins, Joost C. Dessing.

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
