## [Decision Letter · Decision Letter 0]

26 Jun 2020

PONE-D-20-11507

Effects of visual occlusion by the wall on goalkeeper performance during free kicks in football

PLOS ONE

Dear Dr. Valkanidis,

Thank you for submitting your manuscript to PLOS ONE. After careful consideration, we feel that it has merit but does not fully meet PLOS ONE’s publication criteria as it currently stands. Therefore, we invite you to submit a revised version of the manuscript that addresses the points raised during the review process.

Among the different comments, I have specifically noted that your "expert" group do not seem to be judged as true experts by the reviewers. I feel this is an important point to correct, which may also have a deeper impact on your interpretation. It would unlikely be enough to now mention them as "skilled (R2), "advanced" or "intermediate" players (please refer to R3 taxonomy figure) depending on their exact level, we probably need to know more on the practical implication if these players are no longer real experts. It is also very surprising to me that you have two groups in your main experiment (Exp. 1) but choose not to compare them. this needs to be better justified, or to be done I think. Please also see all the comments to address all the points raised by the reviewers.

We look forward to receiving your revised manuscript.

Kind regards,

Robin Baurès, Ph.D.

Academic Editor

PLOS ONE

Journal Requirements:

We note that one or more of the authors are employed by a commercial company: INCISIV Ltd.

2.1. Please provide an amended Funding Statement declaring this commercial affiliation, as well as a statement regarding the Role of Funders in your study. If the funding organization did not play a role in the study design, data collection and analysis, decision to publish, or preparation of the manuscript and only provided financial support in the form of authors' salaries and/or research materials, please review your statements relating to the author contributions, and ensure you have specifically and accurately indicated the role(s) that these authors had in your study. You can update author roles in the Author Contributions section of the online submission form.

2.2. Please also provide an updated Competing Interests Statement declaring this commercial affiliation along with any other relevant declarations relating to employment, consultancy, patents, products in development, or marketed products, etc. 

3. Please include captions for your Supporting Information files at the end of your manuscript, and update any in-text citations to match accordingly. Please see our Supporting Information guidelines for more

Reviewers' comments:

Reviewer's Responses to Questions

**Comments to the Author**

1. Is the manuscript technically sound, and do the data support the conclusions?

Reviewer #1: Yes

Reviewer #2: Partly

Reviewer #3: Partly

2. Has the statistical analysis been performed appropriately and rigorously? 

Reviewer #1: Yes

Reviewer #2: No

Reviewer #3: Yes

3. Have the authors made all data underlying the findings in their manuscript fully available?

Reviewer #1: Yes

Reviewer #2: Yes

Reviewer #3: Yes

4. Is the manuscript presented in an intelligible fashion and written in standard English?

Reviewer #1: Yes

Reviewer #2: Yes

Reviewer #3: Yes

5. Review Comments to the Author

Reviewer #1: The study presents a novel and appropriate use of VR simulator to investigate the strategic implications of a conventional sports strategy: Goalkeepers blocking free kicks taken over a defensive wall.

The technical specifications of the VR set up and calibration is described in exact and replicable detail. However, I would greatly appreciate links to Video 1 (line 141) and Video 2 (152).

The number and arrangement of trials to accommodate variables of ball placement and kick type/trajectory are very thorough. Although I didn’t follow the comment on reducing from 360 to 180 trials (197). “We only analyzed the experimental trials.”

Reviewer instructions are to focus on research design and statistical analysis, and I do have a few points of clarification or revision. One is that the designation of “Novice Goalkeepers” is not defined very thoroughly (line 91). Are they regular amateur players (mean age 27 years)? Related to that (311), the authors state that, “No explicit group comparison was conducted.” This calls for further explanation and justification as readers expect such comparisons in expert-novice studies.

Although the authors refer to statistical significance in the Success Rate within expert or novice groups (31; 478; 514), I would appreciate Table 1 indicating where differences are statistically significant... if only to “thin the heard” of results. Typically, expert-novice research paradigms give weight to differences that reveal expert advantage.

The key variables in this study seem to be the placement of the ball and the duration/height of ball flight. The authors note that the free-kick modeling required kicks to be higher in order to be presented as slower. This should perhaps be echoed a few times through the analysis, since different free-kick takers can certainly generate greater pace without increasing trajectory.

This leads to a question that might be addressed at the end under Future Considerations. While the present study thoroughly investigates the effect of a Wall on goalkeepers, it doesn’t address the effect of a wall on free-kick takers. Some of them, freed of the constraint of kicking over/around the wall, may kick with pace and trajectory that are more challenging for goalkeepers.

Although reviewer instructions do not ask about interest or implications, I am compelled to note that this study establishes a framework for using VR simulators to investigate and challenge conventional strategies in sports. This has great practical value in coaching and player development, and can feed questions back into basic research as well.

When I look at well-conceived and executed research, I look in particular for if and how the “representative tasks” devised by researchers might be re-purposed for testing and training. This study certainly provides methods to test and train goalkeepers in VR.

Reviewer #2: This paper investigated an interesting topic; visual control of action in football goalkeeping during a free kick with and without a wall. Whilst the paper is generally well written, several sections can be improved as follows. First, a clearer theoretical framework is needed of how early and late visual information maps to guide actions with links to relevant analyses. Second, the manuscript is lengthy, difficult to follow in the disjointed presentation of experiments 2 and 3. In addition, I was not convinced by the use of novice participants in experiments 2 and 3. I would limit the paper to experiment 1. Third, participants need to be accurately categorised as skilled, not experts. Fourth, use of VR technology as reported in the methods is impressive, but I am confused by the omission of kicker kinematics. Fifth, the authors should conduct further work using truly expert goalkeepers before they recommend omission of the wall during free kicks. The wall can influence the kicker’s strategy and difficulty of shot execution. Finally, please add a limitations section based upon my comments. More specific details are below. See attached document.

Reviewer #3: This manuscript sets out to assess the effects of visual occlusion by the wall on goalkeeping performance from free kicks which is practically relevant as the free-kicks provide a goal scoring opportunity. The use of an immersive, interactive virtual reality system in the current study allows for a significant degree of experimental control. Please see attachment for reviewer comments.

6. PLOS authors have the option to publish the peer review history of their article (what does this mean?). If published, this will include your full peer review and any attached files.

Reviewer #1: **Yes: **Peter J. Fadde

Reviewer #2: No

Reviewer #3: **Yes: **Alan Dunton

---

## [Author Response · Author response to Decision Letter 0]

16 Oct 2020

All responses to the reviewers were incorporated in the “Response to Reviewers.docx” document. We would like to thank you for your suggestions.

---

## [Decision Letter · Decision Letter 1]

19 Nov 2020

A goalkeeper’s performance in stopping free kicks reduces when the defensive wall blocks their initial view of the ball

PONE-D-20-11507R1

Dear Dr. Valkanidis,

We’re pleased to inform you that your manuscript has been judged scientifically suitable for publication and will be formally accepted for publication once it meets all outstanding technical requirements. Please note that I do not feel skilled enough regarding the question of Prof. Craig's position in Incisiv, if this should be mentioned in the article beyond the affiliation. I have asked a senior editor, more experience than I am on this point, to have a look to this specific point. He might therefore contact you to recommend some action on this specific point.

Kind regards,

Robin Baurès, Ph.D.

Academic Editor

PLOS ONE

Additional Editor Comments (optional):

Reviewers' comments:

Reviewer's Responses to Questions

**Comments to the Author**

1. If the authors have adequately addressed your comments raised in a previous round of review and you feel that this manuscript is now acceptable for publication, you may indicate that here to bypass the “Comments to the Author” section, enter your conflict of interest statement in the “Confidential to Editor” section, and submit your "Accept" recommendation.

Reviewer #1: All comments have been addressed

Reviewer #2: All comments have been addressed

Reviewer #3: All comments have been addressed

2. Is the manuscript technically sound, and do the data support the conclusions?

Reviewer #1: Yes

Reviewer #2: Yes

Reviewer #3: Yes

3. Has the statistical analysis been performed appropriately and rigorously? 

Reviewer #1: Yes

Reviewer #2: Yes

Reviewer #3: Yes

4. Have the authors made all data underlying the findings in their manuscript fully available?

Reviewer #1: Yes

Reviewer #2: Yes

Reviewer #3: Yes

5. Is the manuscript presented in an intelligible fashion and written in standard English?

Reviewer #1: Yes

Reviewer #2: Yes

Reviewer #3: Yes

6. Review Comments to the Author

Reviewer #1: (No Response)

Reviewer #2: Thank you for revising the manuscript. I look forward to reading future work with VR that includes expert goalkeepers and where kinematic information of the kicker is presented.

Reviewer #3: I believe the authors have done well to address the Reviewer comments.

I would like to see an acknowledgement of the potential for goalkeepers to utilise differing control strategies in terms of initial timing and control.

7. PLOS authors have the option to publish the peer review history of their article (what does this mean?). If published, this will include your full peer review and any attached files.

Reviewer #1: **Yes: **Peter J. Fadde

Reviewer #2: No

Reviewer #3: **Yes: **Alan Dunton

---

## [Editor Report · Acceptance letter]

1 Dec 2020

PONE-D-20-11507R1 

A goalkeeper’s performance in stopping free kicks reduces when the defensive wall blocks their initial view of the ball 

Dear Dr. Valkanidis:

I'm pleased to inform you that your manuscript has been deemed suitable for publication in PLOS ONE. Congratulations! Your manuscript is now with our production department. 

Kind regards, 

on behalf of

Dr. Robin Baurès 

Academic Editor

PLOS ONE